# Does Parental Reflective Functioning Mediate the Associations between the Maternal Antenatal and Postnatal Bond with the Child in a Community Sample?

**DOI:** 10.3390/ijerph19126957

**Published:** 2022-06-07

**Authors:** Chiara Pazzagli, Livia Buratta, Giulia Cenci, Elena Coletti, Maria Luisa Giuliani, Claudia Mazzeschi

**Affiliations:** Department of Philosophy, Social Sciences and Education, University of Perugia, 06123 Perugia, Italy; livia.buratta@unipg.it (L.B.); giulia.cenci@unipg.it (G.C.); coletti.elena93@gmail.com (E.C.); marialuisa.giuliani@unipg.it (M.L.G.); claudia.mazzeschi@unipg.it (C.M.)

**Keywords:** maternal–fetal attachment, maternal postnatal attachment, perinatality, parent-to-infant bonding, parental reflective functioning, PRFQ, mediation analyses, prospective study

## Abstract

Although establishing an affective tie with a child during perinatality is considered one of the most important maternal tasks, little is still known about the mediators of the association between maternal antenatal and postnatal bonding with the infant. This prospective study addresses this gap by evaluating a community sample of 110 Italian women to assess whether maternal pre- and postnatal bonds with the infant are mediated by parental reflective functioning (PRF), as assessed at the third trimester of pregnancy and three months postpartum. Controlling for confounding variables, the hierarchical regression analyses show the maternal prenatal quality of attachment to the fetus as the main predictor of maternal postnatal attachment to the child (β = 0.315; t = 0.2.86; *p* = 0.005). The mediation analyses show that mothers’ PRF (b = 0.245; SE = 0.119; 95% CI = 0.071, 0.531) explains 39% of the relationship between maternal pre- and postnatal bonding with the child. The findings of this study contribute to research on the association between prenatal and mother-to-infant bonding by additionally investigating the importance of taking into account maternal PRF as a mediating variable. This provides support for the clinical utility of interventions focused on maternal PRF.

## 1. Introduction

During the perinatal period, the establishment of an affective tie with the fetus and subsequently with the newborn baby is one of the most important maternal tasks and has a predictive role on children’s socio-emotional, behavioral, and cognitive development in early childhood and on the parents’ mental health during perinatality [1,2,3,4,5,6,7,8].

Initially, the theoretical literature which stressed that the relationship between mother and child begins during pregnancy, was mainly of psychodynamic orientation [9,10]. Maternal care behaviors are considered to be grounded in the sensitivity and empathic capacity of the mother who immediately, after the birth, is able to identify and meet the needs of the newborn baby, as suggested by Winnicott [11]. The expecting mother develops an intensive awareness about the physical and psychological needs of her infant throughout the pregnancy, when women frequently enter into an intensely heightened phase of emotional preparedness for parenting [12]. The maternal–fetal tie during pregnancy guides the mother’s thoughts, feelings, and behavior toward her child after the baby is born [13,14]. Although the relationship between the mother and child is a process that begins at least during pregnancy, to date research has mainly focused on postpartum mother–infant interactions [15,16]. Recent reviews that have explored the associations between pre- and postnatal bonding have highlighted that the complex pathways likely to exist in these associations are still unclear [2,15,17]. Considering other factors, such as parental mentalization, could provide a deeper understanding of the variables that influence the establishment of the maternal affective tie with the fetus and subsequently with the newborn [17,18,19].

### 1.1. Background

The mother–fetus tie has often been labeled as “maternal–fetal attachment”, the emotional bond that the pregnant mother establishes with the unborn child. Even though the construct of the maternal–fetal relationship and related tools in this area of research usually refer to “attachment”, various scholars have considered the terms “bond” or “relationship” more appropriate than “attachment”, the latter being a misleading term concerning care-seeking rather than caregiving [3,20]. Walsh [20] defines prenatal attachment as a multi-faceted construct guided by the caregiving system, the complementary parental side of the attachment system, the main function of which is to provide protection, comfort, and care for a child [21]. We refer here to mother–fetus bonding (MFB) when addressing the general issues of maternal prenatal attachment and its specific measures in empirical research.

MFB is grounded on caregiving-based concepts, such as the concept advanced by Condon [22]. Suggesting a model which links the maternal subjective experience with her behavior, for Condon, the maternal bond to the fetus and to the child is driven by the mother’s disposition to know, interact with, to avoid separation or loss, to protect and meet the needs of her child. Such dispositions may (or may not) be translated into overt behaviors such as information seeking, proximity, protection, pleasing, and gratifying. To measure the construct, Condon developed instruments for both the prenatal and postnatal periods [22,23]. Consistent with this perspective, the items of the Maternal Antenatal Attachment Scale (MAAS) focus on conscious thoughts, feelings, behaviors, and attitudes towards the fetus [3,24]. Specifically, the tool assesses the quality of maternal affective experiences towards the unborn child (such as feelings of closeness and tenderness versus feelings of detachment and distance or irritation) and the intensity of preoccupation with the fetus (the strength of feelings toward the unborn child and the amount of time thinking or dreaming about and talking to the fetus). In regard to the postpartum period, the Maternal Postnatal Attachment Scale (MPAS) questionnaire was also designed with the aim of exclusively addressing the parental side of the relationship with the child and assessing the mother’s emotional response to her infant along a number of dimensions focused predominantly on affective responses, rather than on beliefs or attitudes [22].

There is a fairly sizable body of literature examining MFB, but the prevalence of research with cross-sectional designs limits the findings on the predictive aspects of MFB, and there are even fewer studies on its consequences [15,25]. Specifically, in studies employing Condon’s MAAS, the data show that the subscale quality was consistently related to maternal mental health (associations with anxiety and depressive symptoms that were explored further), whereas the intensity subscale was not [15]. At a theoretical level, these findings support the need to consider the MFB as a complex and multidimensional construct, and at an empirical level, the importance of taking into account subscale scores and not interpreting global MFB scores in isolation when conducting research [24,26,27]. Thus, the MAAS subscale scores are considered in the present study.

Recent reviews have explored the associations between pre- and postnatal bonding [2,15,17]. Overall, prospective studies show that MFB is a precursor of postnatal bonding in the early postpartum period. Studies assessing MFB in the prepartum period with MAAS show a positive correlation with MPAS in the postpartum period [28,29,30,31,32,33,34]. Despite these findings, when it comes to the support for patterns of continuity across the antenatal and postnatal periods, research data in some studies indicate only low or moderate associations between pre- and postpartum periods. Considering that the assessed construct is multidimensional and thus can be influenced by other perinatal variables, there are still critical gaps in knowledge about the associations between pre- and postnatal bonding, and studies that take into account intervening variables are necessary, such as the parental mentalization capacities [15,17,30]. From a psychodynamic perspective, empirical studies show that maternal caregiving behaviors are closely related to the maternal mentalization (operationalized as reflective functioning), namely the caregiver’s capacity to reflect upon her own internal mental experiences as well as those of the child [35]. It has been postulated that parental reflective functioning (PRF) allows mothers to create both a physical and psychological experience of comfort and safety for her child [36,37,38]. As Alvarez-Monjarás et al. [19] recently showed, it is the development of a mother’s capacity for reflecting upon the child’s mental states that allows the parent to display more appropriate and sensitive responses to the child’s behavioral cues. Therefore, higher maternal PRF is related to more positive maternal caregiving behaviors [18,19,38]. However, the link between caregiving experiences and maternal PRF has not been thoroughly studied. The Parental Reflective Functioning Questionnaire (PRFQ; Ref. [36]) is a brief multidimensional self-reporting instrument designed to assess PRF capacities. Among the findings of studies that have assessed PRF using the PRFQ, the parent’s capacity to mentalize is a strong predictor of the quality of the parent–child relationship, is related to parents’ ability to provide sensitive care, results in a sense of confidence about parenting, and leads to a parent’s perception of their own ability to cope with parenting [36,39,40].

Since mothers’ PRF is connected to caregiving behaviors, considering the latter as an indicator of reflective functioning [41], we postulated that PRF should be closely related to MFB and to postnatal maternal bonding with the child. Studies investigating the relationship between PRF and MAAS and MPAS are very few. To assess MFB, Røhder and colleagues [42] used MAAS and PRF with an adaptation of PRFQ for use during pregnancy in an at-risk sample, showing a positive correlation between prenatal PRF and both the quality and intensity subscales of MAAS. Høifødt and colleagues [43] used PRFQ and MPAS in order to assess the mother–infant relationship in a study aimed to evaluate the efficacy of a preventive intervention. To our knowledge, no study has explored the postulated association and mediating role of PRF between pre- and postnatal bonding with the child.

### 1.2. Purpose of This Study

Starting from the critical gaps in knowledge about the associations between prenatal and mother-to-infant bonding, the main aim of the present prospective study is to explore, in a community sample, the mediating role played by mothers’ PRF in order to investigate the mechanisms through which prenatal and mother-to-infant bonds are connected. Considering the above-mentioned literature, three specific hypotheses are tested. First, MFB assessed with MAAS is expected to correlate significantly and positively with maternal postnatal emotional bonding, which is in turn assessed with MPAS. MFB is also expected to correlate significantly and positively with higher maternal PRF, particularly with regard to MAAS’ quality subscale. Furthermore, higher maternal PRF is expected to correlate significantly and positively with MPAS. Second, after taking into account mothers’ socio-demographic, obstetric, and mental health variables which previous studies have shown to have a strong relationship with the MAAS [4,44], MAAS subscales are expected to predict MPAS. Specifically, factors that have repeatedly found by the literature to have a positive interaction with the building, processing, and expression of the emotional bond between mother and fetus, have been taken into account, including: lower gestational age [45] Siddiqui et al., 1999), high socio-economic status in terms of educational level [46], having a stable relationship with partner [45], a planned pregnancy [44], and lower levels of depression and anxiety symptoms [4]. Finally, maternal PRF is expected to partially mediate the relationship between prenatal and mother-to-infant bonding. Figure 1 illustrates the conceptual model testing the PRFQ as a mediator between MAAS subscales and MPAS.

## 2. Materials and Methods

### 2.1. Study Design, Sampling, and Setting

In this prospective cohort study the participants took part in two surveys at two different time points, at T0 during the third trimester of pregnancy to assess the mother-to-infant bond with MAAS and at T1, three months after childbirth to assess mothers’ PRF using the PRFQ and maternal postnatal emotional bonding assessed with MPAS. An a priori power analysis was conducted using G* [47] to test a multiple regression using a medium effect size (f^2^ = 0.15), an alpha of 0.05, and five main predictors. Results showed that a total sample of 92 participants was needed to achieve a power of 0.80. Considering an expected missing data rate of 15% [48] and an expected drop-out rate of 25% [49], a convenience sampling of a community sample of 152 future mothers were recruited from different public and private birth centers of the Umbria Region (Italy) between 2016 and 2019 by following these inclusion criteria: (a) woman in the third trimester of pregnancy, who (b) did not have an at-risk pregnancy, and (c) understood the Italian language.

After signing a written informed consent form and agreeing to participate, all participants filled in a paper–pencil booklet of some standardized Italian versions of self-report questionnaires. The recording was carried out at the birth centers during the accompanying course at birth, while the second survey at T1 took place at the women’s homes. At T0, in addition to signing the informed consent form and filling in the booklet, the future mothers also gave consent to be recontacted for the second phase of the study (T1). In this study, only women who completed all the questionnaires at T0 and T1 were included.

The study was conducted in compliance with the ethical standards for research outlined in the Ethical Principles of Psychologists and the Code of Conduct of the American Psychological Association [50].

### 2.2. Participants

Of the 152 questionnaires distributed at T0, 16 were excluded because they were incomplete, and 12 women did not give the consent to be recontacted for the second phase of the study. Of the 124 future mothers who completed the questionnaires at T0, 14 women during T1 refused to continue participating after being recontacted. The final sample of this study consisted of 110 women with a mean age of 32.70 (SD = 4.34; Min 19–Max 48).

### 2.3. Measures

The standardized self-report measures administrated to assess the key constructs of this study were all translated into Italian following the back-translation procedure according to the guidelines developed by the International Test Commission [51] to reduce various bias that can affect the adequacy of instruments and the data interpretation.

#### 2.3.1. Demographic and Mental Health Form

Demographic and mental health information was collected at T0 and included maternal age, educational level, marital status, and the length of their romantic relationships, whether the pregnancy was planned and whether the mother was the parent of other children.

#### 2.3.2. Edinburgh Postnatal Depression Scale (EPDS)

A self-reported questionnaire composed of 10 items that assesses with a four-point scale the depression symptoms in perinatal period [52]. In this study the Italian version, translated and validated by Benvenuti and colleagues [53], has been used. The Italian validation showed a good internal consistency (Cronbach alpha = 0.78), high sensitivity and good positive predictive value. A score of 8/9 has been identified as cut-off in the community screenings. In our study the EPDS showed a Cronbach alpha of 0.75 that highlighted a good reliability, in line with the cut-off recently suggested by Griethuijsen et al. [54].

#### 2.3.3. State and Trait Anxiety Inventory (STAI-Y)

A self-reported questionnaire consisting of 40 items which assesses, using a four-point Likert scale ranging from 1 (not at all) to 4 (very much so), two different kinds of anxiety; the State Anxiety or anxiety about a specific event and the Trait Anxiety as a personal and stable feature. Total scores ranged from 20 to 80 for each scale, with a score of 40 as predictive threshold value of anxiety symptoms [55]. In this study, the Italian version by Pedrabissi and Santinello [56] was administered showing a good internal consistency and adequate test-retest reliability, similarly in line with the original version. In our study the two subscales showed good reliability, respectively, State Anxiety with α of 0.90 and Trait Anxiety with α of 0.82.

#### 2.3.4. Maternal Antenatal Attachment Scale (MAAS)

A self-reported questionnaire composed of 19 items which assesses the mother’s prenatal attachment [24]. The 19 items are scored with a five-point scale and are divided into two dimensions: (1) the quality of attachment (QA), where higher scores indicate a higher quality of the parents’ affective experience towards the unborn child; and (2) the intensity of preoccupation (IP), where higher scores measure a higher intensity of preoccupation with the fetus. In general, high scores for both quality and intensity and for total score indicate a more adaptive MFB. At T0, the Italian version of MAAS was administered [57]. The Italian translated and validated study of MAAS [57] confirmed the bi-dimensional structure of the questionnaire established by the original version [24] and its predictive validity with respect to a measure of postnatal mother–child attachment. About the internal consistency Busonera et al., [57] showed a Cronbach alpha between good (α of 0.71) and acceptable (α of 0.62) for the MAAS total score and IP, respectively, while showing a poor internal consistency for QA (α of 0.57). In our study, the MAAS subscales used showed a Cronbach alpha between acceptable (IP = 0.63) and good (QA = 0.75).

#### 2.3.5. Parental Reflective Functioning Questionnaire (PRFQ)

A self-reported questionnaire composed of 18 items that assesses the multidimensionality of the PRF using a seven-point Likert scale from 1 (strongly disagree) to 7 (strongly agree) [36]. The items are divided into three subscales: (1) Pre-Mentalizing (PM) to capture non-mentalizing modes. Higher scores in this subscale indicate a more non-mentalizing stance; (2) Certainty About Mental States (CMS), concerning the ability to recognize that mental states are not transparent. A higher score in this subscale indicates the ability to recognize that children’s feelings, thoughts, and intentions are not always readily apparent; and finally (3) Interest and Curiosity (IC) to assess the interest and curiosity a parent has in their child’s mental states. Adequate PRF is reflected by low PM and medium-to-high CMS and IC [58]. At T1, a modified Italian version of the PRFQ was administered [59]. Pazzagli et al. in their Italian translated and validated study showed an adequate dimensional structure of the PRFQ in Italian parents. The Cronbach alpha for the three subscales for mothers was 0.51 for IC with an increase to 0.62 after item 14 was excluded; 0.61 for PM and 0.78 for CM. In our study two problematic items (11 and 14) were removed to increase the subscales’ reliability, as previously suggested in the Italian validation study [54]. Finally, the reliability coefficients of our study ranged from acceptable (PM = 0.60; IC = 0.60) to good (CMS = 0.87).

#### 2.3.6. Maternal Postnatal Attachment Scale (MPAS)

A self-reported questionnaire composed of 19 items that assesses the mother’s emotional response to her infant relating to the parent-to-infant bond [23]. Each item of the Italian version administered in this study [60] at T1 scored from 1 to 5, where 1 represents lower bonding and 5 higher bonding. The factor analysis in the Italian translated and validated study did not confirm the three-factor structure of the original version, suggesting to use the Total Score (TOT-MPAS) of the scale. The internal consistency of the TOT-MPAS is adequate (Cronbach’s alpha = 0.77) and the score distribution tends to be skewed towards higher attachment scores. The total score (TOT-MPAS) is calculated within a range from 19 to 95, where low scores correspond to a probable poor relationship with the child. In our study, the TOT-MPAS showed a good reliability (Cronbach α = 0.80).

### 2.4. Data Analysis

Before analyzing the data, subjects who had missing data were deleted. Due to the non-normality distribution of three out of six of the main studied variables (MAAS-QA; PRFQ-PM and PRFQ-IC), non-parametric statistics were performed.

To assess the aims of this study, after running the descriptive statistics in terms of mean and standard deviation and percentage for all analyzed variables, Spearman correlations, hierarchical regression analyses, and mediation analyses were performed.

For the first aim, the Spearman correlation analyses were used to explore the existing relationship among the main variable studied, MAAS subscales, PRFQ subscales and MPAS. The effect sizes were interpreted according to Cohen [56], where 0.10, 0.30, and 0.50 represented small, medium, and strong effects, respectively. Only the correlations between medium and strong were considered.

Subsequently, to verify the direct effects of MAAS subscales on MPAS, controlling for seven potentially confounding variables, the hierarchical regression analyses were run. To examine the significant direct effects and its stability within a larger simulated sample, the bootstrapping technique using 10,000 samples was used. Specifically, in the first block, the main independent variables studied and MAAS subscales were included, whereas demographic (maternal age, education, duration of the romantic relationship, planned pregnancy) and mental health (depressive symptoms, state and trait anxiety symptoms) variables were added in the second and third block, respectively. Information regarding a woman’s marital status and whether or not she was primiparous was excluded due to little variability in their distribution.

Finally, to assess the third aim focused on the investigation of the indirect effects of the MAAS subscale on the MPAS through PRFQ subscales, a mediation analysis was conducted. To control their effects, the seven potentially confounding variables inserted in the hierarchical regression models were also included as covariates in the mediation analysis.

All analyses were carried out with Statistical Package for Social Science version 26, IBM, Armonk, NY, USA [61]. To perform the mediation analysis, PROCESS macro version 4.0 in SPSS, released 21 August 2021, developed by Preacher et al. [62] was used. To examine the significance of the indirect effects, the bootstrapping technique (10,000 bootstrap samples) was selected [63,64]. The mean indirect estimation effects for all mediational paths were computed, as well as the 95% confidence intervals and standard errors for each of these estimates. If zero is not included within the range of the confidence intervals, it is suggested that the indirect effect is significant [64].

## 3. Results

### 3.1. Descriptive Statistics for Demographic and Mental Health Information

A large proportion of the group had a medium/high educational level and 100% of the participants were married or cohabiting with the father of the child, mostly within a longtime relationship. Most of these women were first-time mothers and had planned the pregnancy. Regarding mental health information, the descriptive statistics showed mean levels of EPDS and STAI lower than the clinical cut-off, highlighting a non-clinical population. Table 1 shows the descriptive statistics in terms of mean, standard deviation, and percentage for demographic and mental health information.

### 3.2. Descriptive Statistics and Spearman Correlations for the Main Studied Variables

The Spearman correlations showed many significant relationships with a medium and strong effect. Specifically, MAAS-QA and MAAS-IC showed a positive correlation, both these MAAS subscales showed a positive correlation with MPAS. MAAS-QA also showed a negative relationship with PRFQ-PM and a positive correlation with PRFQ-CMS and PRFQ-IC, while MAAS-IP showed a positive correlation with PRFQ-IC. Regarding the correlations among the PRFQ subscales and MPAS, the PM subscale showed a negative correlation, whereas the CMS subscale was found to be positively correlated. Table 2 reports the means, standard deviations, and correlations for the main studied variables. Highlighted in bold are the correlations between mean and strong that were considered in the subsequent analyses.

### 3.3. Hierarchical Regression Models with MAAS Subscales as Predictors of MPAS

The data showed an increase in the explained variance between the three models, form 27% for the first block to 36% for the third block, all showing a significant effect with a *p* < 0.001. Specifically, with regard to the effect of MAAS on MPAS, only the MAAS-QA subscale was a significant predictor of MPAS in all three blocks (*p* < 0.01), with a *p* = 0.005 in the third block where all seven confounder variables were inserted. The MAAS-QA subscale in all three blocks showed a stronger β coefficient than all other variables, proving to be the best predictor of MPAS even after adding the demographics and mental health variables. On the contrary, the MAAS-IP subscale did not reach significance in any block. For this reason, the MAAS-IP was no longer taken into account in the subsequent analyses. Considering the increase in the explained variance with the addition of the confounding variables in the second and third blocks, these variables were also taken into account in the subsequent analyses. Table 3 reports the three hierarchical regression models tested to verify the effects of MAAS subscales on MPAS to clarify the effect of the confounding variables. Highlighted in bold are the regression effects between mean and strong of the variables studied and considered in the subsequent analyses.

### 3.4. Mediation Analyses: PRF as Mediator

Figure 2 illustrates the mediation analyses testing the PRFQ subscales as a mediator between MAAS-QA and MPAS. PRFQ-PM and PRFQ-CMS were two of three PRFQ subscales which showed significant correlations with both MAAS-QA and MPAS. The data highlighted a significant total effect of the model tested (R^2^ = 0.48; F _(10,97)_ = 8.91; *p* < 0.001). The overall model explained 48% of the MPAS’ variance.

MAAS-QA and the two PRFQ subscales were the only significant variables of the overall model with *p* < 0.05, as shown in Table 4. A significant direct effect of the MAAS-QA (b = 0.624; 95% CI = 0.246, 1.00) on the MPAS was found. The direct effect of MAAS-QA on the MPAS was partially mediated by both PRFQ subscales. The total indirect effect was b = 0.245 (SE = 0.119; 95% CI = 0.071, 0.531). Specifically, the MAAS-QA had a significant indirect effect on MPAS through PRFQ-PM (b = 0.140; SE = 0.089; 95% CI = 0.015, 0.354) and through PRFQ-CMS (b = 0.105; SE = 0.063; 95% CI = 0.013, 0.257).

## 4. Discussion

The main aim of the present study was to examine the extent to which prenatal and postnatal maternal bonds with a child are mediated by mothers’ PRF. In order to investigate this, three hypotheses have been formulated. Consistent with the first hypothesis, results from this community sample suggest that MAAS subscales are significantly and positively correlated with maternal postpartum bonding. This matches with previous findings, suggesting the important role of the conscious emotional tie experienced by mothers in pregnancy towards the fetus on the subsequent maternal affective response to her infant [28,29,30,31,32,33]. Furthermore, still consistent with the first hypothesis, an association between MFB and maternal PRF emerged from the study. Specifically, only taking into account correlations with medium to strong effects, the data show a stronger effect in the correlation between MAAS’ quality subscale and PRFQ subscales than MAAS’ intensity of preoccupation subscale. The difference in the strength of the correlations with PRFQ between the two MAAS subscales is in line with previous studies, showing that the intensity dimension is more affected by environmental factors (such as age and employment), whereas the quality dimension is more affected by maternal mental health issues (for a recent review, see McNamara et al. [15]). Overall, the data indicate that a mother’s increased experience of closeness in the relationship with the fetus (MAAS quality dimension) is negatively related to the extent to which she reported struggling to understand and interpret her child’s mental states (PM subscale), and is positively related to the extent to which she stated she was certain about her children’s mental states (CMS subscale). The results of the same two PRFQ subscales significantly correlated with postnatal mother-to-infant bonding. That is, PM has negative correlations with the mother’s emotional response to her infant, whereas CMS has positive correlations with her subjective experience of her overt behaviors oriented toward the child.

Overall, our findings are partially consistent with previous studies suggesting that mothers with an inability to understand her child’s mental state also have less satisfaction with her parenting, less involvement and communication with the child, and a lower sense of efficacy as mothers, while a mother’s perception that her thoughts about the child’s mental states are accurate is related to feelings of satisfaction, parental competence, and her ability to cope with parenting [39,40]. This latter finding is consistent with suggestions that a caregiver’s CMS could reflect maternal confidence in her knowledge about the child, which may result from greater communication and involvement [39,59,65]. Contrary to expectations, the IC subscale was found to have a smaller than medium correlation with both MAAS subscales and a not significant correlation with MPAS. These findings might suggest that this mode of parental mentalization could be differently related, or less relevant, to the maternal subjective experience of her overt (conscious) behaviors oriented toward the fetus and the child, as assessed by MAAS and MPAS. As many items on the MPAS assess the degree (e.g., frequency) of a mother’s subjective experience of enjoying caring for the child (such as the desire for proximity and the enjoyment of interaction; the desire to identify and gratify the infant’s emotional state; a sense of confidence, competence and satisfaction at being mother), the absence of a correlation between the IC subscale and the MPAS can be understood as being partially in line with findings of a previous study showing a positive correlation of the IC subscale with maternal efficacy, but not with satisfaction [41]. Furthermore, it could be speculated that a high correlation between the CMS (and not IC) subscale and the MPAS may be related to the fact that mothers of our study were all in the first postpartum trimester. In the scientific literature, the first postnatal trimester is referred to as the “fourth trimester”, indicating that the mother’s capacity to be intensely aware of, and concerned about, her baby’ needs, both physical and psychological, develops during pregnancy and continues into the first few months of the newborn baby’s life. Thus, the homogeneity of the early age of the infant of our sample, an aspect in which our study differs from the majority of previous research, could be related to the finding of a strong relationship between a mother’s perception that her thoughts about the child’s mental states are accurate (CMS) and her perceived emotional bond toward the fetus. This would be in line with considerations on a possible change in the main caregiver’s reflective capacity during the different stages of early childhood [66,67]. However, at the moment, this is only a speculation based on the theoretical literature and these findings are yet to be fully understood. As no study has investigated the association between PRFQ and MPAS in the first postnatal period, future research is needed to better comprehend this issue.

In regard to the second hypothesis, after taking into account demographic and mental health variables as potentially confounding variables, results from this sample suggest that only the MAAS-QA subscale was a significant predictor of MPAS. Thus, this shows how postnatal mother-to-infant bonding is strongly predicted by the quality of the maternal subjective experience of her overt caregiving behaviors oriented toward the fetus. As the MAAS quality subscale reflects an inner disposition during pregnancy to become affectively involved with her unborn child with feelings of closeness and tenderness [43], the data show that, when the effects of the confounding variables are controlled, this disposition in turn predicts the mother’s MPAS in terms of greater pleasure in proximity, acceptance, tolerance, and competence as a parent [23].

Finally, we assumed that maternal PRF could mediate the relationship between prenatal and mother-to-infant bonding (third hypothesis). Based on the consideration that MAAS and MPAS aim to investigate the maternal subjective experience of her overt (conscious) behaviors oriented toward the fetus and the child, whereas PRFQ assesses a mental disposition that assists the mother in understanding her child’s emotions, thoughts, and behaviors guiding the caregiving system into action [35,41,68,69], we hypothesized that maternal PRF could play a mediating role in the relationship between her affective responses towards the fetus and towards the child. The results indicate that 39% of the effect of MAAS’ quality subscale on MPAS is explained by the PM and CMS scales. These effects were observed when all confounding variables (e.g., demographics and mental health information) were included as covariates. Thus, maternal feelings of closeness and tenderness during pregnancy toward the fetus (high MFB quality) partially affect a mother’s acknowledgment that her thoughts about her child’s mental states are accurate postnatally (CMS subscale). This in turn influences a mother’s affective responses experienced when caring for the infant, indicating a high emotional bond (high MPAS). On the contrary, maternal feelings of detachment and distance or irritation with the fetus during pregnancy (low MFB quality) have a partial effect on the mother’s reported struggle to understand and interpret her child’s mental states postnatally (PM subscale), which in turn influences the maternal experience of a low emotional bond toward the infant (low MPAS).

These findings support psychodynamic theories suggesting that maternal care behaviors are grounded in the capacity of the mother who, thanks to a state of mind defined by Winnicott [11] as “primary maternal preoccupation”, is able to identify with the infant and to meet the needs of the newborn baby. Our findings seem to indicate that the association between the experience of a pregnant woman when thinking about the fetus, and her subjective experiences of the bond toward the infant is mediated by the maternal PRF, thus “capacity to envision her child as motivated by internal mental states such as feelings, wishes, and desires” [70] (p. 175).

To the best of our knowledge, this is the first study to explore PRF as a mediator between prenatal and postnatal maternal bonding with a child. The study is exploratory in nature, and the findings need to be replicated before firm conclusions can be drawn. The current study has other limitations, including the fact that only self-reporting measures were used, as individuals may offer biased estimates that can range from a misunderstanding of what a correct measurement is to social desirability bias [71]. Larger prospective and multi-method studies in this field are encouraged, which would enable a more in-depth analysis of the assessed associations. Thirdly, the study involves only a community sample and future studies need to collect data also for at risk first-time mothers. Furthermore, as the main focus was on three constructs (i.e., MFB, maternal PRF, and mother to infant bond), despite having controlled for the effect of other variables as possible confounding variables (i.e., socio-demographic, obstetric and mental health variables) on the basis of previous studies, there are other potential influences that have not been included in the statistical model. Including additional measurements in future studies could solve this limitation.

## 5. Conclusions

Overall, the present study suggests that maternal PRF is one of the mechanisms through which the maternal subjective experience with her behavior in the relationship with the fetus is associated with the emotional bond experienced towards the newborn child, even when including a relevant number of confounding variables. Considering that critical gaps still remain in the knowledge about the complex pathways that are likely to exist between pre- and postnatal maternal bonding, the present study sheds some light on the variables that mediate this relationship in the perinatal period. Given the importance of the early mother–child relationship, the findings of this study have important implications for clinical practice by indicating the important mediating role of PRF in strengthening or improving the maternal emotional bond experienced toward the infant, and by stressing the importance of considering pregnancy as a crucial period in the development of the capacity to meet the needs of the newborn baby. Furthermore, our data confirm the findings of previous studies that postulated PRF to be connected to caregiving behaviors, considering the latter an indicator of reflective functioning. The mother’s capacity (or difficulties) for reflecting upon the child’s mental states promotes (or hinders) the maternal emotional response to caregiving behaviors postnatally.

These preliminary findings have potentially important consequences both for understanding the complex pathways between pre-and postnatal maternal bonding, and, by highlighting the mediating role played by PRF, providing possible directions on how to support mothers in strengthening or enhancing emotional bonding with the child effectively.

## Figures and Tables

**Figure 1 ijerph-19-06957-f001:**
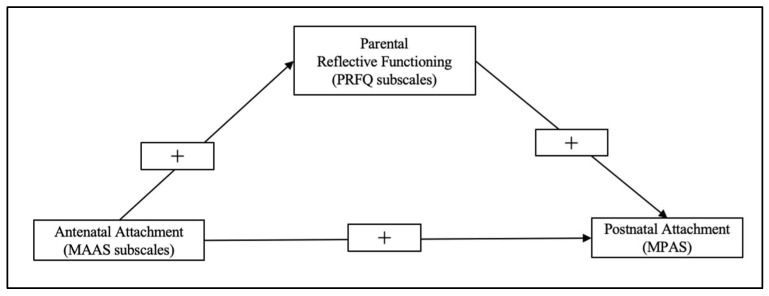
The conceptual model.

**Figure 2 ijerph-19-06957-f002:**
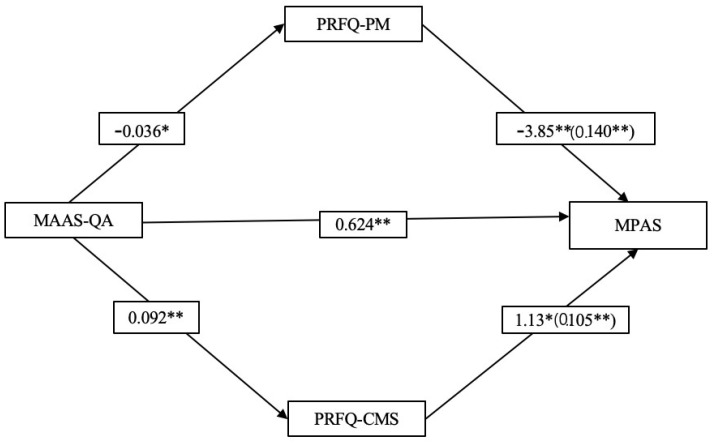
Mediation analysis testing the direct and indirect effects of MAAS-QA on MPAS mediated by PRFQ subscales. * *p* < 0.05, ** *p* < 0.01.

**Table 1 ijerph-19-06957-t001:** Descriptive statistics of demographic and mental health variables (N 110).

Demographic Variables	N	%
**Educational level**		
Upper secondary school	28	25.5
Up to secondary school	82	74.5
**Marital status**		
Married or Cohabiting	110	100
**Duration of the romantic relationship**		
Less than 5 years	29	26.4
5 years or more	81	73.6
**Parenting experience**		
First-time mother (primiparous)	98	89.1
One or more previous children	12	10.9
**Planned pregnancy**		
Yes	83	76.1
No	27	23.9
	**Mn**	**SD**
**Pregnancy week**	32.44	3.62
**Mental health variables**	**Mn**	**SD**
EPDS	7.47	3.87
STAI State	37.14	8.49
STAI Trait	38.74	6.67

Notes: EPDS = Edinburgh Postnatal Depression Scale; STAI = State and Trait Anxiety Inventory.

**Table 2 ijerph-19-06957-t002:** Descriptive Statistics and Spearman correlations among the main studied variables (N 110).

	Mn	SD	(2)	(3)	(4)	(5)	(6)
(1) MAAS-QA	46.76	3.11	**0.473 ****	**−0.393 ****	**0.375 ****	0.286 **	**0.569 ****
(2) MAAS-IP	25.51	3.62	----	−0.125	0.160	0.279 **	**0.302 ****
(3) PRFQ-PM	1.50	0.56	----	----	----	----	**−0.485 ****
(4) PRFQ-CMS	4.41	1.17	----	----	----	----	**0.433 ****
(5) PRFQ-IC	6.28	0.64	----	----	----	----	0.185
(6) MPAS	87.20	7.00	----	----	----	----	----

Notes: MAAS = Maternal Antenatal Attachment Scale; QA = Quality of Attachment; IP = Intensity of Preoccupation; PRFQ = Parental Reflective Functioning Questionnaire; PM = Pre-Mentalizing; CMS = Certainty About Mental States; IC = Interest and Curiosity; MPAS = Maternal Postnatal Attachment Scale. ** *p* < 0.01.

**Table 3 ijerph-19-06957-t003:** Hierarchical multiple regression models and MAAS subscales’ effects on MPAS, controlling for the confounding variables.

Dependent Variable: MPAS										
Model	Predictor	B (95% CI)	*SE*	*Β*	*T*	*p*	*R^2^*	*F*	*gdl*	*p*
**1**							0.269	19.287	2, 105	0.000
	MAAS-QA	1.14 (0.709; 1.65)	0.239	**0.507**	5.08	0.000				
	MAAS-IP	0.040 (−0.256; 0.421)	0.200	0.020	0.205	0.838				
**2**							0.282	6.61	6, 101	0.000
	MAAS-QA	1.08 (0.601; 1.61)	0.254	**0.479**	4.64	0.000				
	MAAS-IP	0.078 (−0.322; 0.479)	0.205	0.041	0.388	0.699				
	Age	−0.032 (−0.277; 0.244)	0.132	−0.020	−0.226	0.822				
	Education	−1.461 (−4.41; 1.83)	1.58	−0.090	−1.029	0.306				
	Relationship	1.48 (−1.13; 4.16)	1.37	0.091	1.04	0.301				
	Pregnancy	0.315 (−2.07; 2.71)	1.21	0.019	0.224	0.823				
**3**							0.359	6.09	9,98	0.000
	MAAS-QA	0.710 (0.175; 1.29)	0.277	**0.315**	2.86	0.005				
	MAAS-IP	0.220 (−0.221; 0.659)	0.220	0.114	1.10	0.276				
	Age	0.016 (−0.241; 0.283)	0.133	0.010	0.109	0.913				
	Education	−1.35 (−4.27; 1.52)	1.46	−0.083	−0.983	0.328				
	Relationship	2.30 (−0.152; 4.72)	1.25	0.142	1.66	0.101				
	Pregnancy	1.05 (−1.56; 3.64)	1.31	0.064	0.759	0.450				
	EPDS	−0.079 (−0.562; 0.347)	0.231	−0.043	−0.384	0.702				
	STAI-State	−0.014 (−0.259; 0.235)	0.125	−0.017	−0.123	0.902				
	STAI-Trait	−0.292 (−0.520; −0.021)	0.126	−0.277	−2.16	0.040				

Notes: MPAS = Maternal Postnatal Attachment Scale; MAAS = Maternal Antenatal Attachment Scale; QA = Quality of Attachment; IP = Intensity of Preoccupation; Education = Educational Level; Relationship = Duration of the Romantic Relationship; Pregnancy = Planned Pregnancy; EPDS = Edinburgh Postnatal Depression Scale; STAI = State and Trait Anxiety Inventory. B confidence intervals were based on 7904 samples.

**Table 4 ijerph-19-06957-t004:** Effects of each variable on MPAS included in the overall model.

Dependent Variable: MPAS					
Predictor	Coeff	SE	t	*p*	95% CI
MAAS-QA	0.624	0.191	3.27	0.001	0.246, 1.00
PRFQ-PM	−3.85	1.09	−3.27	0.001	−6.04, −1.67
PRFQ-CMS	1.14	0.554	2.05	0.043	0.036, 2.24
Age	0.047	0.134	0.349	0.728	−0.219, 0.312
Education	−1.18	1.25	−0.943	0.348	−3.66, 1.30
Relationship	2.18	1.26	1.73	0.087	−0.324, 4.69
Pregnancy	−0.935	1.29	−0.725	0.470	−3.49, 1.62
EPDS	0.041	0.188	0.218	0.828	−0.332, 0.415
STAI-State	−0.093	0.106	−0.877	0.382	−0.303, 0.117
STAI-Trait	−0.130	0.127	−1.02	0.309	−0.383, 0.123

Notes: MPAS = Maternal Postnatal Attachment Scale; MAAS = Maternal Antenatal Attachment Scale; QA = Quality of Attachment; PRFQ = Parental Reflective Functioning Questionnaire; PM = Pre-Mentalizing; CMS = Certainty About Mental States; Education = Educational Level; Relationship = Duration of the Romantic Relationship; Pregnancy = Planned Pregnancy; EPDS = Edinburgh Postnatal Depression Scale; STAI = State and Trait Anxiety Inventory. Confidence intervals were based on 10,000 samples.

## Data Availability

The data that support the findings of this study are available on request from the corresponding author.

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
