# Peer review of "Does Parental Reflective Functioning Mediate the Associations between the Maternal Antenatal and Postnatal Bond with the Child in a Community Sample?"

_ijerph, 2022, doi:10.3390/ijerph19126957_

Round 1
Reviewer 1 Report
Thank you for the opportunity to review the manuscript, Does parental reflective functioning mediate the associations between the maternal antenatal and postnatal bond with the child in a community sample? submitted to the International Journal of Environmental Research and Public Health. Overall, I found the study to be well designed with some limitations that can be addressed with revisions. The manuscript requires revisions, refinements, and editing to present clear, concise, and clean statements. The form and flow was difficult to follow due to the density of the introduction, and the lack of some essential reporting elements in the methods section. Specific recommendations for improvement are provided in the attached marked pdf and the review summary below. These recommendations are intended to help the authors successfully revise the manuscript for publication. For this reason, the recommendations and feedback are critical but constructive. Once the content is easier to follow and the limitations are addressed by the authors, this manuscript will make a nice contribution to the literature. Thanks again for the opportunity to review this good work.
REVIEW SUMMARY (also please see the attached pdf for more detailed comments).
INTRODUCTION - This section is too dense with a lack of concept clarity. There is too much information about the instruments in this section that should be presented in the methods section as a subsection in data collection called instruments. This will be detailed in the next section of the review summary. As this is a health sciences rather than a psychology journal, I would suggest organizing the section into Introduction and then another section called Background.
Introduction (section)
-----First three paragraphs to introduce the topic in terms of the general focus of the manuscript as well as the central concept of interest.
Background (section)
-----One paragraph to summarize the MAJOR concepts/constructs that will be discussed in the subsection called Conceptual Model (or Framework). Then, outline each concept/construct with a heading followed by a clear and concise description or supporting details. Please refrain from focusing on the instruments, instead, focus on the concept/construct that will be measured. The instruments should be presented in the methods section to operationalize the concept/construct in the data collection. Finally, close the background with the last paragraph to present the purpose of the research.
Attachment (subsection of conceptual model)
Caregiving behaviors (subsection of conceptual model)
Purpose (subsection of background)
Finally, I highly recommend providing a diagram or figure for the introduction that clearly states the hypotheses with arrows for relationships and +/- for the expected pathway. Let me add, this is an important diagram as there is not clarity about the mediators as opposed to moderators. This diagram can then be used to detail the findings more clearly in the results.
Methods (section)
Study Design (subsection of methods)
-----Please state the study design
Setting and Sample (subsection of methods)
-----There are multiple standard reporting elements missing. For example, the sampling technique was not stated, and there was no power analysis provided about the ideal sample size. See the recommended reporting element checklist referenced at the end of this summary for more details.
Data Collection (subsection of methods)
-----Information about the data collection process followed by the description and psychometric properties of the ALL the instruments (see the next subsections). Also, there is information missing in this section such as the dates (month, year to month, year) of data collection. Again, see the recommended reporting element checklist referenced at the end of this summary for more details.
Edinburgh Postnatal Depression Scale (subsection of data collection)
Italian version of the State and Trait Anxiety Inventory (subsection of data collection)
Maternal Antenatal Attachment Scale (subsection of data collection)
Parental Reflective Functioning Questionnaire (subsection of data collection)
Maternal Postnatal Attachment Scale (subsection of data collection)
As a note, the current information provided about the instruments is not sufficient for the psychometric properties. The information from not only the original English language instrument needs to be presented but also the psychometric properties of the instruments used in Italian studies needs to be provided. There were not citations for the Italian studies where the psychometric properties were established, and more importantly the studies from which the instrument translation was completed.
Presenting the Cronbach's alpha is alone not sufficient for demonstrating the instruments are measuring the concepts. This is especially important as the instruments were developed in English but used in this study in Italian. A low alpha value, for example, could be due to a low number of questions, poor inter-relatedness between items, or heterogeneous constructs.
Data Analysis (subsection of methods)
-----Please organize and refine this section with a clear description from descriptive statistics, inferential statistics, and modeling. The flow is difficult to follow as presented and some important information is missing. For example, there is no information about how missing data was managed for analysis. Also, please provide the appropriate statistical references for each step, and specialized technique. For example, the bootstrapping technique should have a citation as well as the how the regression was completed.
RESULTS
-----Usually the overview of the descriptive statistics is first presented followed by the reference to the table for further information. The marital status should be separated into married and not-married but cohabitating. This is relevant to the conceptual model.
DISCUSSION
-----This section could be better referenced in some places, redundancies eliminated with content from the introduction, and revised to be clearer and more concise. Some of the current sentences are difficult to read as they are too long with multiple concepts. The limitations section needs some additional notations about other concerns specific to the biases introduced by this type of study design.
CONCLUSION
-----The conclusion is repetitive with more discussion. There should be no citations in the conclusion. In reality, the last paragraph of the discussion could actually be a much better conclusion.
REPORTING RECOMMENDATIONS
As previously noted in the review summary, please apply the checklist for the cross-sectional studies (https://www.equator-network.org/reporting-guidelines/strobe/) to improve the reporting of the study. The checklist is part of the Strengthening the Reporting of Observational Studies in Epidemiology (STROBE) Statement: Guidelines for reporting observational studies (https://www.strobe-statement.org/). I recommend sending the completed checklist as a supplemental file with page and line numbers indicated for each reporting criteria stated.

Author Response
-
We thank the reviewer for the careful and constructive suggestions. We have reorganised the introduction to make it more focused and flowing. The part on method was also revised in depth and parts of the conclusions were modified. Some parts that were too dense have been revised. We hope that the article is now more fluent and clearer. We have provided a point-by-point response, and you will find our answers in red and italics.
REVIEW SUMMARY (also please see the attached pdf for more detailed comments).
-
Thanks for the suggestions. The terminology to indicate the bond between the mother and the foetus/infant has been revised to make it consistent throughout the text. The country of origin of the sample has been inserted and the information regarding the age of the mothers removed, as suggested. The term maternal task has been left as it is, since it is frequently used in specific scientific literature, starting with Reva Rubin in 1979. Keywords have not been edited following the MeSH terms, Medical Subject Headings, but following the journal's instructions, which state that "We recommend that the keywords are specific to the article, but reasonably common in the discipline in question".
INTRODUCTION - This section is too dense with a lack of concept clarity. There is too much information about the instruments in this section that should be presented in the methods section as a subsection in data collection called instruments. This will be detailed in the next section of the review summary. As this is a health sciences rather than a psychology journal, I would suggest organizing the section into Introduction and then another section called Background.
Introduction (section)
-----First three paragraphs to introduce the topic in terms of the general focus of the manuscript as well as the central concept of interest.
Background (section)
-----One paragraph to summarize the MAJOR concepts/constructs that will be discussed in the subsection called Conceptual Model (or Framework). Then, outline each concept/construct with a heading followed by a clear and concise description or supporting details. Please refrain from focusing on the instruments, instead, focus on the concept/construct that will be measured. The instruments should be presented in the methods section to operationalize the concept/construct in the data collection. Finally, close the background with the last paragraph to present the purpose of the research.
Attachment (subsection of conceptual model)
Caregiving behaviors (subsection of conceptual model)
Purpose (subsection of background)
Finally, I highly recommend providing a diagram or figure for the introduction that clearly states the hypotheses with arrows for relationships and +/- for the expected pathway. Let me add, this is an important diagram as there is not clarity about the mediators as opposed to moderators. This diagram can then be used to detail the findings more clearly in the results.
-
Thank you for your input. In the revised version of the article, we have taken up the suggestion to divide the introduction into several sections. In the new version we have divided the section into Introduction (pp 1-2; lines 27-48), Background (pp 2-3; lines 50-127), and Purpose of this study (p 3 – 4; lines (130-160) .
-
Regarding the observation on the distinction between the caregiving and attachment systems, as we reported in the revised version at the beginning of the background section, starting from Walsh's work, recent literature on the subject agrees that the attachment, as assessed by Condon instruments, concern the caregiving system, the complementary parental side of the attachment system, the main function of which is to provide protection, comfort, and care for a child.
-
Regarding the presence of the description of the tools in this section of the article, although we agree that it should generally only be in the methods section, we decided to leave a brief description of Condon’s instruments as it allows for a more-in-depth explanation of the specific construct assessed. The latter, in fact, show how the specific construct is multidimensional, an important aspect for explaining the need to investigate mediators, which is the aim of this study. In addition, previous research shows that the two sub-scales of the MAAS predict different aspects in postpartum. The brief description of the two subscales in the introduction section allows us both to report the results of these studies and to formulate the hypotheses of our study in more detail. Finally, we have added a diagram explaining the rationale of the study, which as suggested will certainly allow the reader to clearly grasp the aims of the study (p 4; lines 158-159)
METHODS (section)
Study Design (subsection of methods)
-----Please state the study design
Setting and Sample (subsection of methods)
-----There are multiple standard reporting elements missing. For example, the sampling technique was not stated, and there was no power analysis provided about the ideal sample size. See the recommended reporting element checklist referenced at the end of this summary for more details.
Data Collection (subsection of methods)
-----Information about the data collection process followed by the description and psychometric properties of the ALL the instruments (see the next subsections). Also, there is information missing in this section such as the dates (month, year to month, year) of data collection. Again, see the recommended reporting element checklist referenced at the end of this summary for more details.
Edinburgh Postnatal Depression Scale (subsection of data collection)
Italian version of the State and Trait Anxiety Inventory (subsection of data collection)
Maternal Antenatal Attachment Scale (subsection of data collection)
Parental Reflective Functioning Questionnaire (subsection of data collection)
Maternal Postnatal Attachment Scale (subsection of data collection)
As a note, the current information provided about the instruments is not sufficient for the psychometric properties. The information from not only the original English language instrument needs to be presented but also the psychometric properties of the instruments used in Italian studies needs to be provided. There were not citations for the Italian studies where the psychometric properties were established, and more importantly the studies from which the instrument translation was completed.
Presenting the Cronbach's alpha is alone not sufficient for demonstrating the instruments are measuring the concepts. This is especially important as the instruments were developed in English but used in this study in Italian. A low alpha value, for example, could be due to a low number of questions, poor inter-relatedness between items, or heterogeneous constructs.
-
About the method section, following the suggestions and STRoBE checklist, we have restructured the method section to make it more fluid for the reader and to improve the manuscript. We have reported some elements that were missing in the previous version of the paper, to better detail the research procedures.
-
We divided the method section into several subsections. In the new version we have divided the section into study design, sampling and setting (p 4; lines 162-184), participants (p 4; lines 185-190), measures (pp 5-6; lines 192-269) and finally data analysis (pp 6-7; lines 272-305).
-
In the measures subsection (pp 5-6; lines 192-269), we have created subsections for each measure used (Italian version of the Edinburgh Postnatal Depression Scale; Italian version of the State and Trait Anxiety Inventory; Italian Version of Maternal Antenatal Attachment Scale; Italian Version of Parental Reflective Functioning Questionnaire; Italian Version of Maternal Postnatal Attachment Scale). Furthermore, more detailed psychometric information for each measure has been added.
Data Analysis (subsection of methods)
-----Please organize and refine this section with a clear description from descriptive statistics, inferential statistics, and modeling. The flow is difficult to follow as presented and some important information is missing. For example, there is no information about how missing data was managed for analysis. Also, please provide the appropriate statistical references for each step, and specialized technique. For example, the bootstrapping technique should have a citation as well as the how the regression was completed.
-
Data analysis subsection has been re-organized adding missing information. The appropriate statistical references for each statistical technique have been provided. Reorganizing the section, should now make it more fluid and clearer (pp 6-7; lines 272-305).
RESULTS
-----Usually, the overview of the descriptive statistics is first presented followed by the reference to the table for further information. The marital status should be separated into married and not-married but cohabitating. This is relevant to the conceptual model.
-
As suggested, in the results section, the reference to the table has been moved to the end of each paragraph.
-
In our conceptual model, we considered to be important, in line with previous literature, as reported in the purpose of the study section, to consider whether the mothers had a partner (being married or cohabiting) or were single. For this reason, we asked the mothers if they had a partner (they were displaced or cohabiting) or if they were single.
DISCUSSION
-----This section could be better referenced in some places, redundancies eliminated with content from the introduction, and revised to be clearer and more concise. Some of the current sentences are difficult to read as they are too long with multiple concepts. The limitations section needs some additional notations about other concerns specific to the biases introduced by this type of study design.
-
Following suggestions, some repetitive parts were removed from the introduction and some sentences were reworded to be clearer. A paragraph of the discussions was moved to the conclusion, following the suggestion (p 13; lines 502-524). The limits were revised, taking into account that the study design is prospective (p 13; lines 487-501).
CONCLUSION
-----The conclusion is repetitive with more discussion. There should be no citations in the conclusion. In reality, the last paragraph of the discussion could actually be a much better conclusion.
-
The conclusion has been profoundly revised following the suggestions. Thanks.
REPORTING RECOMMENDATIONS
As previously noted in the review summary, please apply the checklist for the cross-sectional studies (https://www.equator-network.org/reporting-guidelines/strobe/) to improve the reporting of the study. The checklist is part of the Strengthening the Reporting of Observational Studies in Epidemiology (STROBE) Statement: Guidelines for reporting observational studies (https://www.strobe-statement.org/). I recommend sending the completed checklist as a supplemental file with page and line numbers indicated for each reporting criteria stated.
-
We attached as a supplementary file a completed checklist with page and line numbers indicated for each reporting criteria stated. "Please see the attachment"
Reviewer 2 Report
Does parental reflective functioning mediate the associations 2 between the maternal antenatal and postnatal bond with the 3 child in a community sample?
This study addresses in a community sample whether maternal pre- and postnatal bonds with a child are mediated by mothers’ parental reflective functioning. The analyses show the maternal prenatal quality of attachment to the fetus as the main predictor of maternal postnatal attachment to the child. The mediation analyses show that mothers’ parental reflective functioning explains the relationship between maternal pre- and postnatal bonding with the child. In general, the paper is well-written and easy to follow. The results are interesting for the public. I only have some minors to note.
Introduction:
- Sentence: “has an impact on important outcomes for both mother and child in the perinatal period and beyond [1-4].” Please add some more information on what kind of outcomes and specify direction.
- Sentence: “The initial academic consideration of the connection between the pregnant woman 32 and her developing fetus was principally psychodynamic in orientation”. This sentence is difficult to understand; please add more clarification what is meant by –principally psychodynamic in orientation--.
- Related to the previous two points, I would rewrite the first part of the introduction. When proceeding with reading the paper, I can see that child and maternal outcomes, as associated with MFB, are more clearly described in sentences 80-88. I would address the associated outcomes only at one place in the Introduction.
- While the introduction is clear at many points, it is somewhat lengthy at places. Please look at spots where you can shorten, especially when topics are addressed that are not directly assessed in this study.
Materials and Methods
- Not all confounders used directly make sense. For example, why is their controlled for the length of romantic relationships? Please include the rationale of including the specific confounders.
- Related to the previous point: as many confounders are included, but the sample size is limited, careful consideration of confounders included is even more important. Maybe not all confounders are needed?
- The procedure and the materials description are intertwined (page 4 and 5): please first describe what is measured when, and then describe the materials.
- Maybe I’m overseeing this, but in Table 2 the correlation between MAAS-QA and MAAS-IP is not visible. I would include it (and if included in the text, I would also include it in the Table 2).
Discussion
- The discussion is in general well-written and clear. I would appreciate some discussion why the subscale PRFQ-IC is not associated with the MPAS.
- Also, the authors speculate that observational methods would be more desirable, but how can you observe feelings of bonding and parental reflective functioning? Please specify.
Author Response
We would like to thank the reviewer the constructive suggestions, which have enabled us to clarify further some aspects. We have provided a point by point response. You will find our answers in red and italic
Introduction:
- Sentence: “has an impact on important outcomes for both mother and child in the perinatal period and beyond [1-4].” Please add some more information on what kind of outcomes and specify direction.
- Thanks for the suggestion, in the revised version we have specified this aspect better (p. 1; lines: 29-31)
- Sentence: “The initial academic consideration of the connection between the pregnant woman 32 and her developing fetus was principally psychodynamic in orientation”. This sentence is difficult to understand; please add more clarification what is meant by –principally psychodynamic in orientation--.
- The sentence has been reworded (p 1; lines: 32-33) “the theoretical literature which stressed that the relationship between mother and child begins during pregnancy, was mainly of psychodynamic orientation [9, 10]”
- Related to the previous two points, I would rewrite the first part of the introduction. When proceeding with reading the paper, I can see that child and maternal outcomes, as associated with MFB, are more clearly described in sentences 80-88. I would address the associated outcomes only at one place in the Introduction.
- Thanks for the suggestion. In the revised version the repetition has been removed.
- While the introduction is clear at many points, it is somewhat lengthy at places. Please look at spots where you can shorten, especially when topics are addressed that are not directly assessed in this study.
- The introduction has been re-organised and divided into three sections, in order to make it more focused and shorter. Thanks
Materials and Methods
- Not all confounders used directly make sense. For example, why is their controlled for the length of romantic relationships? Please include the rationale of including the specific confounders.
- Related to the previous point: as many confounders are included, but the sample size is limited, careful consideration of confounders included is even more important. Maybe not all confounders are needed?
- We considered important taking into account the confounding variables, which literature about this topic has repeatedly been shown to be strongly correlated with maternal prenatal attachment, as also added in the “purpose of the study” section (p 3; lines 140-147). To take into account the confounding variables, a methodology used in previous studies has been followed (i.e. Nordhal et al., 2020)
- The procedure and the materials description are intertwined (page 4 and 5): please first describe what is measured when, and then describe the materials.
- We followed your suggestions and have first described what is measured when (p 4; lines 164-165), followed by the description of the materials (pp 5-6; lines 192-269).
- Maybe I’m overseeing this, but in Table 2 the correlation between MAAS-QA and MAAS-IP is not visible. I would include it (and if included in the text, I would also include it in the Table 2).
- We added both in Table 2 and in the text, the correlation between MAAS-QA and MAAS-IP (p 8; line 323).
Discussion
- The discussion is in general well-written and clear. I would appreciate some discussion why the subscale PRFQ-IC is not associated with the MPAS.
- Hypotheses about the possible reasons why IC subscale does not correlate with MPAS were explored further on p.12 (lines 430-448). As this is the first study to explore the associations between MAAS, PRFQ and MPAS, the hypotheses proposed are based on theoretical literature and these results will need to be replicated to be fully understood.
- Also, the authors speculate that observational methods would be more desirable, but how can you observe feelings of bonding and parental reflective functioning? Please specify.
- We have reformulated the limits to be more precise. Now, we have reported that: Larger prospective and multi-method studies in this field are encouraged, which would enable a more in-depth analysis of the assessed associations (p 13, lines 487-501)
Reviewer 3 Report
This is an exceptionally well written and rigorous paper. Thank you for the contribution. Your findings reveal intricacies of maternal-fetal/infant bonding.
Consider adding to your identified limitations the fact that the demographics indicate privilege in many variables. Therefore, the degree of bonding might also be a function of privilege.
Author Response
This is an exceptionally well written and rigorous paper. Thank you for the contribution. Your findings reveal intricacies of maternal-fetal/infant bonding.
Consider adding to your identified limitations the fact that the demographics indicate privilege in many variables. Therefore, the degree of bonding might also be a function of privilege.
We thank you for these words which encourage us in our work. We have added it as a limit: the study involves only a community sample and future studies need to collect data also for at risk first-time mothers sample (p 13, lines 487-501).